# Brief communication: Evaluation of the ESA CCI+ ESMR v1.1 sea-ice concentration product

Stefan Kern

Integrated Climate Data Center, Center for Earth System Research and Sustainability, University of Hamburg, Hamburg, 20144, Germany

*Correspondence to*: Stefan Kern (stefan.kern@uni-hamburg.de)

**Abstract.** I evaluated a novel NIMBUS-5 Electrically Scanning Microwave Radiometer (ESMR) sea-ice concentration (SIC) data product. 50 Landsat-1 Multispectral Scanner (MSS) images obtained in the Northern Hemisphere during 1974 were manually classified into open water and ice, mapped onto the ESMR product's grid (25 km resolution) and used to compute Landsat-1 SIC. The resulting ~3300 grid cells, covering mostly compact sea ice, have a mean difference (median), standard deviation, and linear correlation coefficient of -1.4% (0.0%), 6.0%, and ~0.9, respectively. This suggests using this novel ESMR SIC data product as an extension of existing SIC climate data records back in time.

## 1 Introduction

The sea ice cover of the polar oceans has been decreasing in the Northern Hemisphere for the past 40+ years (e.g. Wang et al., 2024) and seems to undergo a regime shift in the Southern Hemisphere (e.g. Purich and Doddridge, 2023). Our knowledge about these changes is to a large extent based on records of the sea-ice concentration (SIC) derived from observations of satellite microwave radiometers. Most of the satellite climate data records (CDRs) of the SIC begin in October 1978 when the Scanning Multichannel Microwave Radiometer (SMMR) on NIMBUS 7 became available (e.g. Lavergne et al., 2019; Meier et al., 2024). Prior to the NIMBUS 7 SMMR sensor there were other satellites carrying microwave radiometers, e.g. the NIMBUS-5 Electrically Scanning Microwave Radiometer (ESMR). That satellite operated in the years 1972 to 1977, potentially providing a valuable extension of SIC records back in time. Recently, Kolbe et al. (2024) published a SIC data product that has been derived from NIMBUS-5 ESMR observations within the European Space Agency (ESA) Climate Change Initiative (CCI) sea ice essential climate variable (ecv) project. Another ESMR SIC data product has been published by the National Snow and Ice Data Center (NSIDC) (Parkinson et al., 2004). In this brief communication, I show results of an evaluation of these two ESMR SIC data products against sea-ice concentration estimates from manually classified Landsat-1 Multispectral Scanner (MSS) imagery.

## 2 ESA CCI+ ESMR sea-ice concentration

The ESA CCI+ NIMBUS 5 ESMR SIC data set, version 1.1, that is evaluated here was obtained from Tonboe et al. (2025) for both hemispheres for the entire period (1972-1977). The data set comes on a 25 km grid resolution EASE2.0 grid. The following variables were used: "ice_conc", "raw_ice_conc_values" – aka the unfiltered, originally retrieved SIC values, "total_standard_error" and "algorithm_standard_error". The "algorithm_standard_error" is the retrieval uncertainty taking into account uncertainties in the brightness temperatures due to instrument and geophysical noise and their correction for the atmospheric influence, tie points and other retrieval-relevant quantities. The "total_standard_error" is the squared sum of the retrieval uncertainty and the uncertainty resulting from the gridding of the SIC values computed at the sensor's footprint scale into a predefined 25 x 25 km² EASE grid – the so-called smearing or resampling uncertainty (see Tonboe et al., 2016; Kolbe et al., 2024).

Monthly values of the sea-ice extent (SIE, the sum of the area of all ice covered grid cells), and the sea-ice area (SIA, the sum of the area of all ice covered grid cells taking into account the actual sea-ice concentration) were computed from monthly mean SIC values. This was done for two SIC thresholds: 15% and 30% to illustrate that the choice of this threshold is crucial, especially in the Southern Hemisphere and more for SIE than SIA; Kolbe et al. (2024) used a threshold of 30%. The monthly mean SIC were computed beforehand from the daily ESMR SIC data. While the ESA CCI+ ESMR SIC data product v1.0 exhibits many days with missing data (Tonboe et al., 2023; Kolbe et al., 2024), the new version v1.1 used here contains substantially less such days. I did not compute monthly means for months with daily data from 12 or fewer days.

The time series of SIA and SIE for the Northern and Southern Hemisphere, shown in Fig. 1 a) and 1 b), respectively,  match well to the results published by Kolbe et al. (2024). Gaps in the time series shown result from periods with missing ESMR data or months with too few daily SIC data – as described above. Comparing the monthly SIE values obtained using the two different SIC thresholds, I find differences between 1 and 1.5 million km² in the Northern Hemisphere and between 1 and 3 million km² in the Southern Hemisphere. Only a part of this difference in SIE naturally results from using the two different thresholds specified. A circum-Antarctic band of 25 x 25 km² grid cells located at 60 degrees latitude with SIC values between 15% and 30% would result in a SIE contribution of only about 500 000 km². At least half of the differences in the SIE computed here using a SIC threshold of 15% of 30% is caused by the relatively large retrieval noise over open water, resulting from a less reliable correction of the atmospheric influence, and a larger uncertainty of the tie points used (Kolbe et al., 2024; Tonboe et al., 2025).

Mean retrieval uncertainties (derived from all grid cells with SIC > 10%) remain smaller than 10% in the Northern Hemisphere (Fig. 1c) but occasionally exceed this value in the Southern Hemisphere (Fig. 1d); overall, retrieval uncertainties tend to be larger than for, e.g. the EUMETSAT / ESA CCI OSI-450 SIC CDR (Lavergne et al., 2019). The mean total (retrieval + sampling, see above) uncertainty peaks during the summer / fall in the respective hemisphere, reaching values > 20% a few times in the Northern and many times in the Southern Hemisphere. Overall, Northern Hemisphere mean total uncertainties tend to be smaller than those in the Southern Hemisphere. This is credible in light of the different geographical settings, which

allows the sea ice cover to be more open for larger areas in the Southern Hemisphere where the sea ice is bounded by open ocean towards the North at every longitude, in contrast to the Northern Hemisphere.

The histograms of the daily retrieval and total uncertainties (Fig. 1 e, f) illustrate that at daily temporal scale the retrieval uncertainties exhibit a dominant mode at around 3% in both hemispheres. The tail towards higher retrieval uncertainty values extends towards higher values in the Southern Hemisphere. Similarly, the total uncertainty exhibits a dominant mode at 6% in both hemispheres. This mode is followed by a relatively weak secondary mode at 14% in the Northern Hemisphere and a very well pronounced secondary mode at 20% in the Southern Hemisphere, in line with the differences in the time series of the monthly mean uncertainties presented in Fig. 1 c) and d). Note that all uncertainties I wrote about in this last paragraph are provided together with the SIC product, resulting from the processing.

## 3 NSIDC ESMR V1 sea-ice concentration

The Nimbus-5 ESMR SIC product was provided by the National Snow and Ice Data Center (NSIDC) from https://nsidc.org/data/nsidc-0009/versions/1 (last access: Sep. 1 2025). A description of the data processing for this product is given in the product users' guide to the data set (Parkinson et al., 2004). In this product, SIC values between 0 and 15% are flagged as low concentration values but can be re-computed as was done for this study to use the same SIC range as for the ESA CCI+ ESMR SIC data product.

## 4 Landsat-1 MSS data

For the evaluation I followed the approach of Kern et al. (2022) using Landsat-1 Multispectral Scanner (MSS) images, converted into surface broadband albedo and subsequently classified into surface types open water, thin/bare sea ice, and thick/snow-covered ice applying an albedo threshold. In total, 284 (Northern Hemisphere: 260; Southern Hemisphere: 24) Landsat-1 MSS images of the Collection 2 Level 1 product were selected and downloaded as L1GS (Level-1 Systematic Corrected) and, when available, L1TP (Level-1 Terrain Corrected) images from the USGS Earthexplorer website (https://earthexplorer.usgs.gov/, last access Sep. 1 2025; download of data: Jan. 27 2023).. These were produced by the Landsat Product Generation System (LPGS) and are available in Cloud Optimized Geographic Tagged Image File Format (geoTIFF) (COG). I performed a quality check, discarding scenes that are too dark, too cloudy or too noisy. A large number of the MSS images exhibits scanlines with missing data or values that are clearly outliers. At the end, I decided to only use Landsat-1 MSS data of the year 1974 because of the best overlap with available ESMR SIC data; in total, I used 50 Landsat-1 MSS scenes. Most of these are from months April and March and all of these are from the Northern Hemisphere. Of the seven Landsat-1 MSS channels only channels 4 to 7 provided useful data; these channels are identical to channels 1 to 4 used by the successors of Landsat-1 (Engebretson, 2020).

In contrast to Kern et al. (2022), I had to use any possible combination of three channels out of the four channels 4 to 7 to derive the broadband surface albedo because at least in one of the channels too many scanlines contained failures. Most often, I used channels 4, 5, and 7. I documented the used combination of channels together with the albedo thresholds for the classification (see below) in a separate metadata file that is provided alongside with the classified Landsat-1 MSS scenes. Metadata required by the pre-processing of the scenes such as the sun elevation angle and viewing angle are provided together with the images. Coefficients for the derivation of the albedo from the reflectance values, such as calibration coefficients, wavelength range information, and mean exo-atmosphere solar radiation, were taken from Chander et al. (2009). Information required for the atmospheric correction was taken from Koepke (1989) and adopted to the Landsat channels used (see Kern et al., 2022). I computed the surface albedo maps and classified them manually into the three above-mentioned surface types at the original resolution of the Landsat-1 MSS images of 60 m. The resulting surface type maps I stored together with a manually derived cloud mask in netCDF file format for further use (Kern, 2025). The entire processing of the Landsat-1 scenes is done with the tool SNAP v9.1 (https://earth.esa.int/eogateway/tools/snap, last access Sep. 1 2025).

The Landsat-1 MSS images classified into open water and sea ice were co-located with the ESMR SIC data following Kern et al. (2022). For every 25 km grid cell of the ESMR SIC data set, I find the respective overlapping Landsat-1 MSS image pixels (60 m resolution). I compute a Landsat-1 SIC at 25 km grid resolution by counting the number of Landsat-1 MSS pixels classified as sea ice that fall into that 25 km grid cell. One such grid cell contains about 173 600 Landsat-1 MSS pixels with 60 m resolution. If less than ten pixels are classified into one surface type, i.e. thick/snow-covered ice, thin/bare ice, or open water, I set the respective fraction to zero. Furthermore, if less than five percent of the Landsat-1 MSS pixels falling into the 25 km grid cell are assigned clear-sky, i.e. less than about 8 680 pixels, I set the respective Landsat-1 SIC to a missing value. The NSIDC ESMR SIC product, which is provided on the NSIDC polar-stereographic grid, was re-projected onto the EASE2 grid using cdo remapnn (https://code.mpimet.mpg.de/projects/cdo, last access Aug. 29 2025).

**5 Intercomparison**

From the 50 classified Landsat-1 MSS images, I obtained about 3300 ESMR SIC grid cells at 25 km grid resolution. The majority of these grid cells are from areas with near-100% SIC – either in the central Arctic Ocean, the Hudson or the Baffin Bay; only few Landsat-1 MSS scenes overlap the ice edge (see Fig. 2 c). Consequently, the comparison is dominated by high SIC cases. This applies to both ESMR SIC data products. For ESA CCI+ ESMR SIC, I find a reasonable distribution of ESMR SIC values binned to Landsat-1 SIC 10% bins around the 1-to-1 line of perfect agreement (Fig. 2b); the squared linear correlation coefficient is close to 0.8, indicating a reasonable agreement. The overall mean difference ESMR minus Landsat SIC is -1.4%; the median difference is 0.0%. Clearly, these low values are the result of the large fraction of near-100% SIC values in both the ESMR and the Landsat-1 SIC data sets. For NSIDC ESMR SIC, I find that low Landsat-1 SIC values are overestimated while high Landsat-1 SIC values are underestimated (Fig. 2 d). A regression would have a slope of 0.6

(compared to 0.8 for ESA CCI+ ESMR SIC) and the linear correlation is rather weak. Most remarkable, however, is the overall large mean underestimation of the Landsat-1 SIC values by 16% (median: 18%) by the NSIDC ESMR SIC data product.

Sea-ice concentration retrieval algorithms tend to saturate near 100% SIC; often an unknown fraction of naturally retrieved SIC values that is larger than 100% is truncated and/or folded back to 100%. This jeopardizes evaluation results like shown in Fig. 2 b) and d) because the true variability of the ESMR SIC values around 100% remains unknown and not considered in the evaluation results. As demonstrated by Kern et al. (2019), the consequence of not taking the natural variability around 100% into account is a too small error (mean difference, here -1.4%) and a too small standard deviation of the mean error (here: 6.0%). For the ESA CCI+ ESMR SIC data product, raw, non-truncated SIC values are available. How these values compare to the Landsat-1 SIC is shown in Fig. 2 a). Clearly, a substantial fraction of the SIC values is higher than 100%. I find a small reduction in the squared linear correlation coefficient but an increase in the slope of the linear fit (0.842 instead of 0.816). The mean error is now 0.7% and the standard deviation is 7.4%, a bit larger than the one obtained using the truncated ESMR SIC values – as expected. This is, overall, a very good agreement between the ESA CCI+ ESMR SIC values and the Landsat-1 SIC.

I also looked at results of an inter-comparison for individual regions. The mean errors (standard deviations) for the central Arctic Ocean, the Hudson Bay, and the Baffin Bay are -0.4% (6.3%), 0.6% (5.8%), and 4.0% (8.2%), respectively, with the majority (2000) data pairs coming from the central Arctic Ocean region. Kern et al. (2022, Table 10) show results of a comparison of OSI-450 SIC values for 28 near-100% SIC cases selected from Landsat-5 images acquired between 2003 and 2011 (see Kern et al., 2022, Fig. 1 a) for their location). For that sub-set, Kern et al. (2022) found mean (median) difference SSM/I OSI-450 SIC minus Landsat-5 SIC of -3.2% (-2.4%) and a standard deviation of the difference of 4.1% for the non-truncated OSI-450 SIC, based on 1978 grid cells. The evaluation results obtained here for the ESMR SIC product agree within their standard deviations with the results of Kern et al. (2022).

One of the main improvements between v1.1 of the ESA CCI+ ESMR SIC CDR and its predecessor version v1.0 (Tonboe et al., 2023; Kolbe et al., 2024), is the treatment of ice types (first-year ice versus multiyear ice) in the retrieval. Fig. 3 exemplifies the differences obtained comparing the non-truncated SIC of both ESA CCI+ ESMR SIC CDR versions with the Landsat-1 SIC estimates. For the Hudson Bay, dominated by a near-100% ice cover of first-year ice, going from v1.0 to v1.1 reduces the mean (median) bias from 11.2% to 0.6% (11.5% to 0.0%), accompanied by a small reduction in the standard deviation of the mean (Fig. 3 a, c). For the central Arctic Ocean (Fig. 3 b, d), going from v1.0 to v1.1 also results in a reduction of the mean (median) bias from 5.3% to -0.4% (2.8% to -0.4%), accompanied by a considerable reduction in the standard deviation of the mean by 4%. Notable is also an improvement in the linear agreement between ESMR and Landsat-1 SIC. I cannot make a definite statement about the exact multiyear ice fraction of the central Arctic Ocean subset used, but it seems very likely that two thirds of the respective Landsat-1 scenes (see Fig. 1 c) are pre-dominantly covered by multiyear ice. With that I conclude that the inclusion of ice types into the ESA CCI+ ESMR SIC retrieval has led to an improvement in the accuracy of the SIC, both for first-year and for multiyear ice.

# 6 Discussion, Conclusions & Outlook

I report about an evaluation of a novel sea-ice concentration data product based on Nimbus-5 ESMR single-channel microwave radiometer observations covering the period December 1972 through May 1977 published in 2024. This product, developed in the framework of the ESA CCI+ sea ice ecv project, is the second one of its kind, complementing a similar data product published by the NSIDC in 2004. For the evaluation, I used 50 Landsat-1 MSS images obtained in the year 1974 in the Northern Hemisphere, mostly during later winter and spring. I computed the surface albedo and manually classified the obtained albedo maps in surface types open water, thin/bare ice, and thick/snow covered ice. By counting ice-covered pixels in the classified Landsat-1 images after their co-location with the ESMR sea-ice concentration data product provided on a 25 km grid resolution EASE grid, I obtained about 3300 25 km grid cells with Landsat-1 sea-ice concentration. Comparing the ESMR with the Landsat-1 sea-ice concentration, I find a convincing agreement for the ESA CCI+ SIC. Both data sets are related linearly with a slope around 0.84, a squared linear correlation coefficient close to 0.8, and a mean difference (median) of -1.4% (0.0%); the standard deviation of the difference is 6.0%. For the NSIDC product I find a considerably weaker linear relationship between both data sets and a mean (median) difference of about -16% (-18%), suggesting a substantial underestimation of the Landsat-1 sea-ice concentration.

The presented evaluation results for the novel ESA CCI+ ESMR sea-ice concentration data product agree well with the results presented in Kern et al. (2022, e.g. their tables 5, 6 and 10). They carried out an inter-comparison between various SIC data sets derived from Special Sensor Microwave/Imager & Sounder and/or Advanced Microwave Scanning Radiometer data and SIC estimated from Landsat-5 and Landsat-8 imagery. Actually, the numbers for the mean and median SIC differences are smaller than those reported by Kern et al. (2022) and the degree of linear correlation is similarly high. Therefore, the maturity of the novel ESMR sea-ice concentration data product evaluated is high enough to reliably extend the SIC data record back in time beyond the NIMBUS 7 SMMR era.

As shown and discussed in Kern et al. (2022), SIC data obtained from Landsat imagery the way done here, has a few deficiencies. One is that the sub-grid scale distribution of sea ice in a Landsat pixel usually causes an over-estimation of the actual SIC. For instance, six sub-grid scale snow-covered sea ice floes of dimension 10 m x 10 m and an albedo of 0.8 distributed in a 60 m by 60 m Landsat-1 MSS image pixel translate in an ice coverage of 1/6 or about 17%. The average albedo of that pixel is, however, 0.17 times 0.8 plus 0.83 times 0.07, equaling about 0.19. This value is above the threshold I chose to separate open water from ice. Therefore, the classification of this pixel results in 100% ice. The degree of this overestimation is a function of the actual sea-ice concentration in the Landsat pixel and of the albedo of the sea ice. The second deficiency is that even using a manually derived cloud mask might still leave a number of Landsat pixels with cloud contamination undetected. Such pixels have a higher albedo than open water and with that contribute to an over-estimation of Landsat SIC. The likelihood for this kind of over-estimation is quite small for the results presented here because the sea-ice cover is quite compact, reflecting winter-time freezing conditions. Undetected cloud shadows are less of a problem because the associated reduction in the surface albedo usually does not change the classification result. Other uncertainty source exist, i.e. the

atmospheric correction I applied (see section 2.2), or uncertainties in the geolocation (these are likely to be larger for Landsat-1 MSS images than for the images of the later Landsat sensors used by Kern et al. (2022)). It is beyond the scope of this brief communication to delve into these sources of uncertainties here.

The limitation to one year (1974) of Landsat-1 MSS images resulted in only 50 classified images available for the inter-
190 comparison. Despite this being a larger Landsat sample than used by previous studies (e.g., Cavalieri et al., 2006; Wiebe et al., 2009; Lu et al., 2018; Zhao et al., 2021), the presented dataset has its limitations and a larger sample size with a better representation in time, regions and ice regimes would be desirable. A useful next step would therefore be to expand the inter-comparison to Landsat-1 MSS images of the other years of the ESMR period.

I do not show results for the Southern Hemisphere here. The main reason is that for the obtained 24 Landsat-1 MSS images
there exist practically no overlap with valid ESA CCI+ ESMR SIC data and the contamination with clouds is considerably worse than in the Northern Hemisphere. Therefore, without further evidence and efforts I cannot make a more credible statement about the quality of that SIC data product other than referring to the consistency in the SIA and SIE time series between the ESMR and the SMMR period (Kolbe et al., 2024) and the report provided along with the data product (Tonboe et al., 2025). Future efforts should include revisiting the Landsat-1 MSS image archive with a focus on the Antarctic to check
whether there are not more images that can be used, and making use of the existing cloud-free parts of the few existing images as much as possible.

Two other general directions of future work include to focussing more on Landsat-1 MSS scenes overlapping the ice edge to obtain more credible evaluation results also at the lower ESMR SIC values. Finally, the results that I showed using the non-truncated ESA CCI+ ESMR SIC values clearly demonstrate one more time that it is very useful, if not even mandatory to carry
out evaluation of sea-ice concentration data products taking the full range of the naturally retrieved SIC values into account. Only in this case the difference between the product and what is considered as the truth or reference, and the standard deviation of this difference, will reveal the full bias.

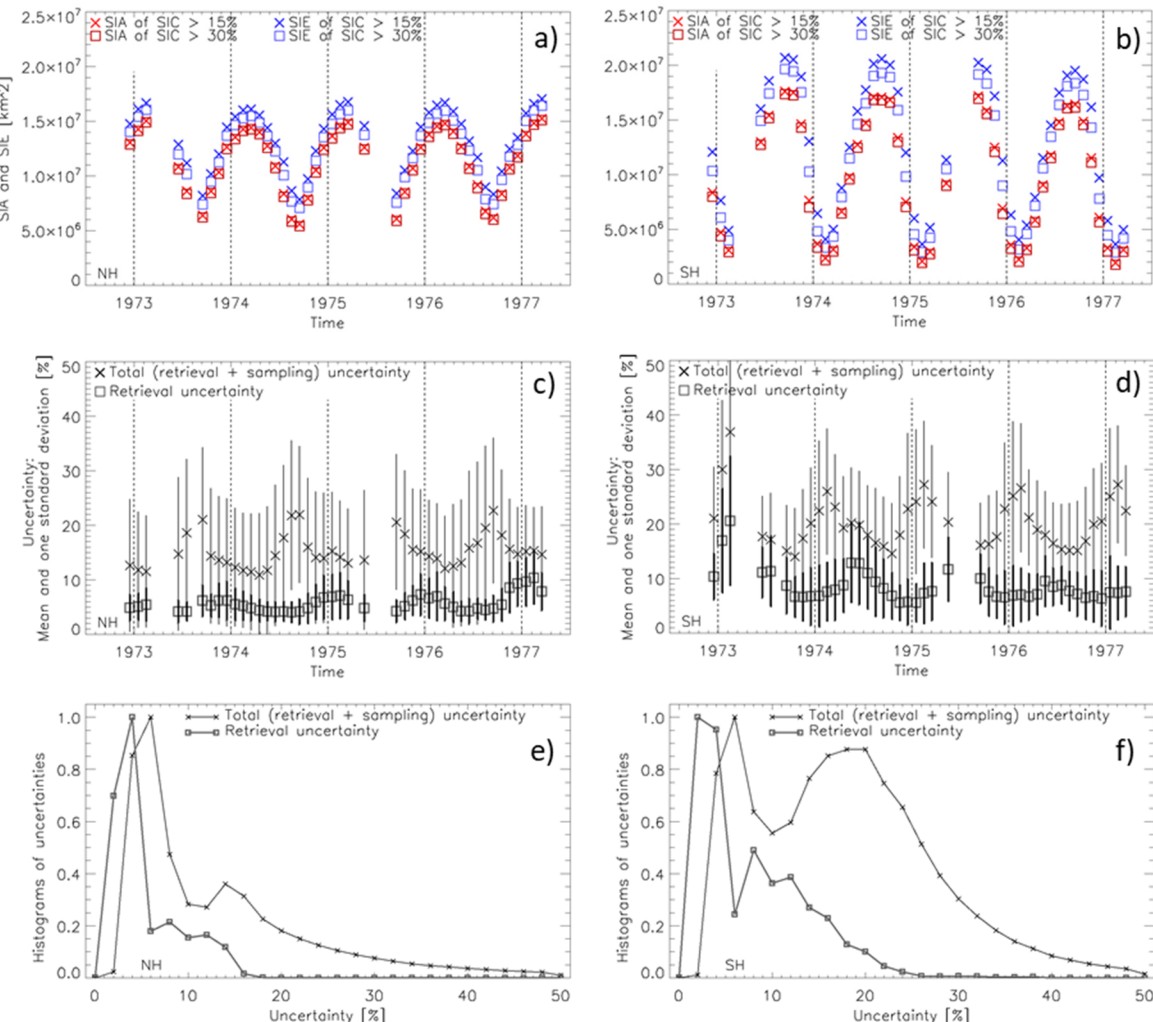

**Figure 1: Time series of the monthly sea-ice area (SIA, displayed in red) and sea-ice extent (SIE, displayed in blue) computed from monthly mean Nimbus 5 ESMR ESA CCI+ v1.1 sea-ice concentration (SIC) (panels a) and b)). Time series of the monthly mean retrieval and total (= retrieval + sampling) uncertainties (panels c) and d)); the bars denote plus/minus one standard deviation of the mean. Histograms of the daily values of the retrieval and total uncertainty using a bin size of 2% (panels e) and f)). Panels on the left hand side are for the Northern Hemisphere, panels on the right hand side for the Southern Hemisphere.**

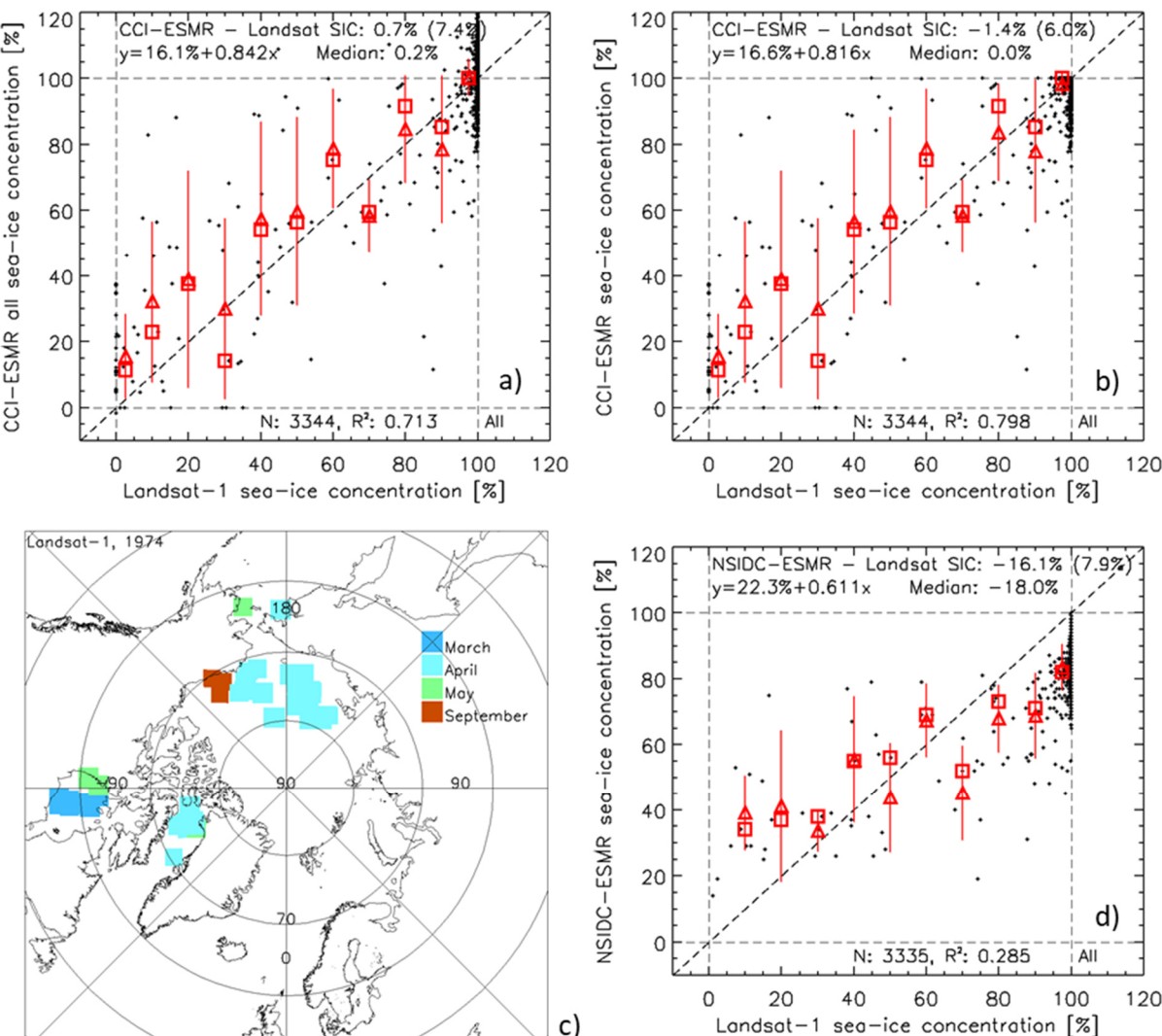

**Figure 2: Scatterplots of the comparison between daily ESMR SIC and Landsat-1 SIC using the non-truncated (a) and truncated (b) ESA CCI+ ESMR SIC product or the NSIDC ESMR SIC product (d). The map in panel c) illustrates the location of the used Landsat-1 scenes together with the month of acquisition in the year 1974. Scatterplots are superposed with the mean-per-bin ESMR SIC (triangles) and with the median-per-bin ESMR SIC (squares); bins used are [0…5[,[5…15[,[15…25[,…,[85…95[,[95…100] Landsat-1 SIC; a minimum of three values is required for the mean or median values. All scatterplots show the mean difference (one standard deviation) in the top right, the equation of the linear fit in the top left and at the bottom, the number of grid cells N and squared value of the linear correlation coefficient R².**

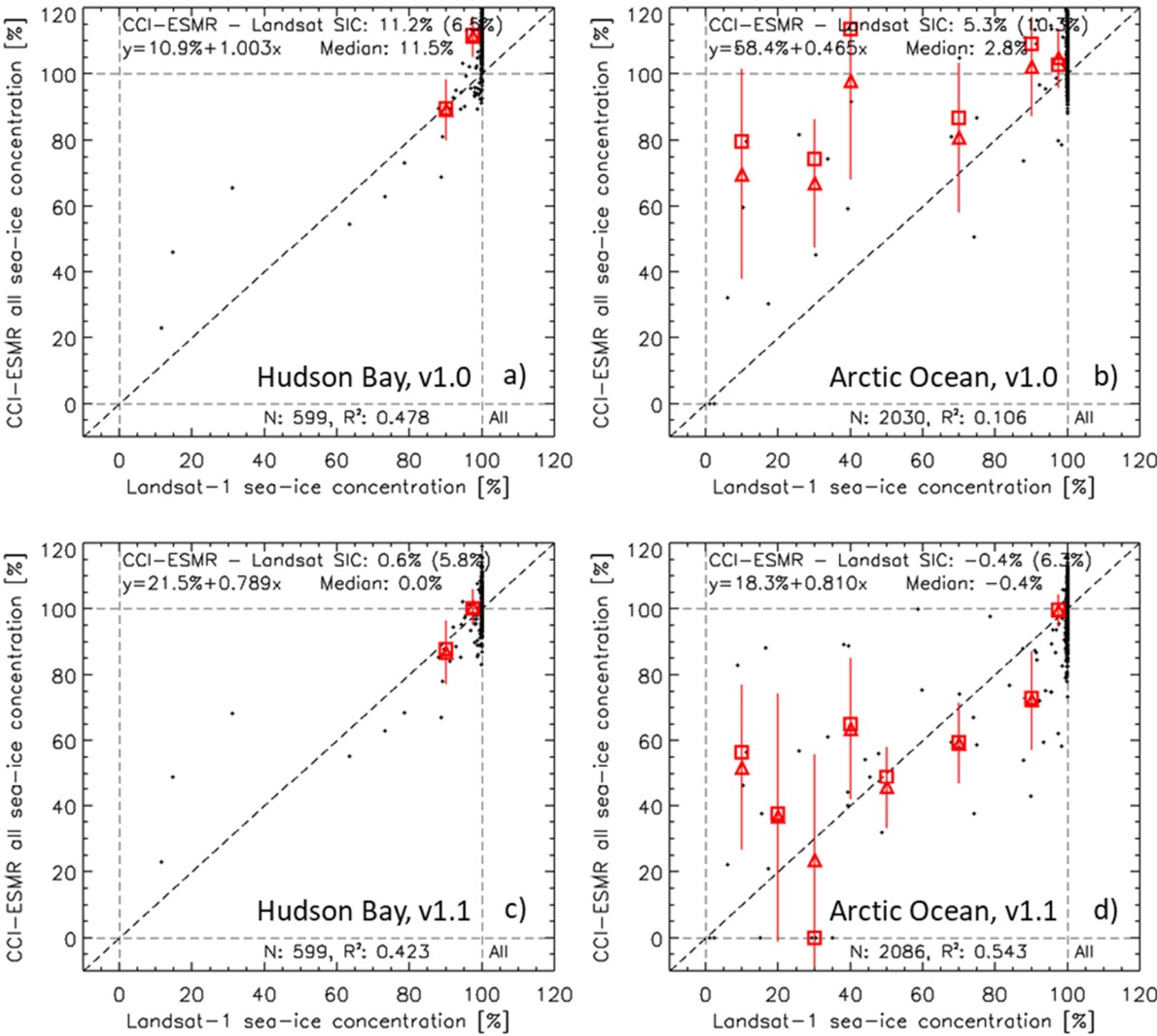

**Figure 3: Scatterplots of the comparison between daily ESMR SIC and Landsat-1 SIC using the non-truncated ESA CCI+ ESMR SIC product v1.0 (a, b) and v1.1 (c, d) for the Hudson Bay (left hand side) and the Arctic Ocean (right hand side). Scatterplots are superposed with the mean-per-bin ESMR SIC (triangles) and with the median-per-bin ESMR SIC (squares); bins used are [0…5[,[5…15[,[15…25[,…,[85…95[,[95…100] Landsat-1 SIC; a minimum of three values is required for the mean or median values. All scatterplots show the mean difference (one standard deviation) in the top right, the equation of the linear fit in the top left and at the bottom, the number of grid cells N and squared value of the linear correlation coefficient R².**

*Data availability.* The classified Landsat images are available from https://doi.org/10.25592/uhhfdm.17915 [last access: Sep. 24, 2025]. The ESMR sea-ice concentration data are available from:

https://doi.org/10.5285/8978580336864f6d8282656d58771b32 [last access: August 29, 2025] and

https://doi.org/10.5067/W2PKTWMTY0TP [last access: August 29, 2025].

*Author contributions.* The author performed the data analysis and wrote the manuscript.

*Competing interests.* The author declares that they have no conflict of interest.

*Acknowledgements.* The publication contributes to the Cluster of Excellence 'CLICCS – Climate, Climatic Change, and Society' and to the Center for Earth System Research and Sustainability (CEN) of the University of Hamburg. The author acknowledges the European Space Agency (ESA) Climate Change Initiative (CCI) program under which umbrella this work has been carried out as part and follow-on of the Sea-Ice ECV CCI+ project.

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
