# Peer review of "Brief communication: Evaluation of the ESA CCI+ ESMR v1.1 sea-ice concentration product"

_EGUsphere, 2025_

## Author Comment (AC1)

Reply to comments of reviewer #1 for manuscript egusphere-2025-5222. Comments of the reviewer are in italic font; responses are given in regular font.

*Review of "Evaluation of the ESA CCI+ ESMR v1.1 sea ice concentration product" by Stefan Kern.*

*Evaluation of sea ice concentration (SIC) is very important for product development and assessment of product quality. There are a number of good reasons to extend the SIC data records back in time, before 1978, and evaluation of these records using independent SIC estimates is particularly important because there is little overlap between SIC products in the 1970s. Stefan has been developing a valuable tool for evaluating SIC products using high resolution optical satellite imagery from Landsat and this has been used for more contemporary SIC records and published (Kern et al. 2022). Here the NIMBUS 5 ESMR v1.1 SIC has been evaluated. I have some suggestions which I think could improve the MS and then some specific comments below.*

Thank you for taking the time to read the manuscript, for the positive perception of its content and for your suggestions to improve the manuscript. I give my response right after each of your suggestions and cite – where appropriate the text changed in and/or added to the manuscript.

*50 Landsat-1 images from the Arctic have been manually classified. The images are mostly from March and April 1974 and obviously the ESMR SIC quality changes seasonally. Could the scenes in Fig. 2c be color coded with time of year so that the reader could get an idea of the seasonal (and geographical) distribution?*

I changed the color of the scenes such that they indicate the month of acquisition.

*One of the updates to ESMR SIC v1.0 to v1.1 was a treatment of the ice types in the SIC processing. The v1.0 SIC was underestimation multiyear ice SIC and overestimating first-year ice SIC. Evaluating this update is important. However, most of the Landsat scenes are over first-year ice and therefore a full evaluation is not possible. Please include a discussion of this limitation in the MS.*

I have added an additional figure (Fig. 3) that shows results of the inter-comparisons of ESMR-v1.0 SIC against Landsat-1 SIC and ESMR-v1.1 SIC against the same Landsat-1 SIC for the subset of the Hudson-Bay and the Arctic Ocean. This figure illustrates that the treatment of ice types realized when going from v1.0 to v1.1 helped reducing the biases between ESMR and Landsat-1 SIC for both first-year ice dominated (Hudson Bay) and multiyear ice dominated (Arctic Ocean) regions. I also added the following text:

"One of the main improvements between v1.1 of the ESA CCI+ ESMR SIC CDR and its predecessor version v1.0 (Tonboe et al., 2023; Kolbe et al., 2024), is the treatment of ice types (first-year ice versus multiyear ice) in the retrieval. Fig. 3 exemplifies the differences obtained comparing the non-truncated SIC of both ESA CCI+ ESMR SIC CDR versions with the Landsat-1 SIC estimates. For the Hudson Bay, dominated by a near-100% ice cover of first-year ice, going from v1.0 to v1.1 reduces the mean (median) bias from 11.2% to 0.6% (11.5% to 0.0%), accompanied by a small reduction in the standard deviation of the mean (Fig. 3 a, c). For the central Arctic Ocean (Fig. 3 b, d), going from v1.0 to v1.1 also results in a reduction of the mean (median) bias from 5.3% to -0.4% (2.8% to -0.4%), accompanied by a

considerable reduction in the standard deviation of the mean by 4%. Notable is also an improvement in the linear agreement between ESMR and Landsat-1 SIC. I cannot make a definite statement about the exact multiyear ice fraction of the central Arctic Ocean subset used, but it seems very likely that two thirds of the respective Landsat-1 scenes (see Fig. 1 c) are pre-dominantly covered by multiyear ice. With that I conclude that the inclusion of ice types into the ESA CCI+ ESMR SIC retrieval has led to an improvement in the accuracy of the SIC, both for first-year and for multiyear ice."

*Overlap between the contemporary SIC records and NIMBUS 5 ESMR is not possible. However, the Kern et al. (2022) article presents a similar evaluation of Landsat imagery and a number of SIC products. The coverage in Hudson Bay and Baffin Bay is comparable to the selection of Landsat in the ESMR evaluation. How does the ESMR evaluation compare with the Kern et al. (2022) evaluation in these two regions?*

Thank you; this is a very good suggestion. The paragraph wherein I reported about the regional results reads now: "I also looked at results of an inter-comparison for individual regions. The mean errors (standard deviations) for the central Arctic Ocean, the Hudson Bay, and the Baffin Bay are -0.4% (6.3%), 0.6% (5.8%), and 4.0% (8.2%), respectively, with the majority (2000) data pairs coming from the central Arctic Ocean region. Kern et al. (2022, Table 10) show results of a comparison of OSI-450 SIC values for 28 near-100% SIC cases selected from Landsat-5 images acquired between 2003 and 2011 (see Kern et al., 2022, Fig. 1 a) for their location). For that sub-set, Kern et al. (2022) found mean (median) difference SSM/I OSI-450 SIC minus Landsat-5 SIC of -3.2% (-2.4%) and a standard deviation of the difference of 4.1% for the non-truncated OSI-450 SIC, based on 1978 grid cells. The evaluation results obtained here for the ESMR SIC product agree within their standard deviations with the results of Kern et al. (2022).

Specific comments:

*In the abstract: Please include the time of year for Landsat scenes and which ice type is covered by the Landsat scenes.*

Since the format "Brief Communication" only allows to use 100 words in the abstract I am bound to keep the abstract as it is. Figure 2 now contains the locations of the Landsat-1 scenes with a color code showing the months (March to May and September). This figure also illustrates that Landsat-1 scenes cover both first-year and multiyear ice. This is now further detailed in the new Figure 3.

*L 14 - 15: include some references to the statement.*

I included two references, one for the Arctic, on for the Antarctic. Please note that the total number of references for a "Brief Communication" is limited to 20; therefore I refrain from putting more references.

*L 15: suggest using "changes" instead of "developments"*

Changed accordingly.

*L 16: add "satellite" before "climate"*

Changed accordingly.

*L 17: delete "first data of the", there was a SMMR on Seasat.*

Changed accordingly.

*L 18: add "on NIMBUS 7" after "(SMMR)" and "NIMBUS 7" before "SMMR"*

Changed accordingly.

*L 18: the sentence starting with "Prior…" rewrite "Prior to the NIMBUS 7 SMMR sensor there were other satellites carrying microwave radiometers, e.g. the NIMBUS 5…."*

Changed accordingly.

*L 31: after "brightness temperatures" add "due to instrument and geophysical noise…"*

Changed accordingly.

*L33: before "EASE" add "a predefined 25 x 25 km2 EASE grid"*

Changed accordingly.

*L37: In Kolbe et al (2024) a threshold of 30% was selected because of the open water noise level. Why 10%? When 15% is more common for contemporary records.*

Thank you, this was an oversight. I changed Figure 1 accordingly, using 15% instead of 10% for the second threshold.

*L40: It is worth mentioning that the coverage in v1.1 is much better than in v1.0 (the list of missing files in v1.1 is much shorter in v1.1).*

Thank you. I added a half-sentence noting this.

*L43: what is meant with "reasonable"?*

I deleted "are reasonable and"

*L45: It is not so clear that it is the SIC thresholds that are compared. Please reformulate.*

The respective part reads now: "Comparing the monthly SIE values obtained using the two different SIC thresholds, I find differences between 1 and 1.5 million km² in the Northern Hemisphere and between 1 and 3 million km² in the Southern Hemisphere. Only a part of this difference in SIE naturally results from using the two different thresholds specified. A circum-Antarctic band of 25 x 25 km² grid cells located at 60 degrees latitude with SIC values between 15% and 30% would result in a SIE contribution of only about 500 000 km². At least half of the differences in the SIE computed here using a SIC threshold of 15% of 30% is caused by the relatively large retrieval noise over open water, resulting from a less reliable correction of the atmospheric influence, and a larger uncertainty of the tie points used (Kolbe et al., 2024; Tonboe et al., 2025)."

*L51: Is this criterion comparable to the evaluation criterion in Kern et al. (2022)? And if not, could this be aligned?*

I am sorry, but I could not follow this comment. I could not find which criterion the reviewer was referring to.

*L85: Is it not better consistently to use 4, 5 and 7 and not just sometimes?*

I wrote in the manuscript that this channel combination was used "most often". There were cases where I could not use this combination of channels because of too many missing or corrupt scanlines in one these channels. As written, I documented the channels that I actually used for the classification of the Landsat-1 images in the metadata of the ice-type maps.

*L95: add "v9.1" after "SNAP" and delete next sentence.*

Changed accordingly.

*L105: Would it not be better to remap the Landsat data onto the Pol. Ster. Grid when evaluating the NSIDC ESMR SIC? So that this process is comparable to remapping the Landsat data to the EASE2 grid when evaluating the ESA ESMER SIC.*

I am following the approach that was used in Kern et al. (2022) as the reviewer has mentioned. The main reason why I chose to remap SIC from the polar-stereographic grid onto the EASE2.0 grid is that I end up with a more consistent set of difference maps between the passive microwave SIC on the one hand and the Landsat SIC on the other hand.

*L131: are these regional differences comparable to Kern et al. (2022)?*

See my answer to the last overarching comment of the reviewer.

*L152: add "NIMBUS 7" before "SMMR"*

Changed accordingly.

*L157: The reasoning about the Landsat SIC systematic uncertainty could perhaps be used for quantifying the Landsat SIC uncertainty including the pixel resolution and SIC dependent uncertainty, albedo variations, instrument noise, atmospheric noise?*

I am not sure I fully understand the comment of the reviewer. I take it as an invitation to dwell deeper into the deficiencies of the Landsat SIC estimates to come up with a per-grid cell uncertainty estimate. However, I could not see in which way I should add this element to the current manuscript.

*L185: last sentence: "will reveal the full bias"?*

Changed accordingly.

---

## Author Comment (AC2)

Reply to comments of reviewer #2 for manuscript egusphere-2025-5222. Comments of the reviewer are in italic font; responses are given in regular font.

*This manuscript investigates two sea ice concentration data sets based on observations of ESMR, the first microwave radiometer in space. As the ESMR data set (1972 – 1977) expands our sea ice data record by 6 years into the past, this quality assessment and comparison is a substantial contribution to our knowledge about features and quality of those data sets.*

Thank you for your positive perception of this manuscript and the results therein. Thank you also for reading the manuscript and providing support to improve the manuscript.

*It is a pity that the comparison with Landsat imagery could not be performed in the Antarctic. It would add more weight to the comparison covering both hemispheres. However, the data at hand in themselves already are worth being published. At least, in the conclusions the extension to the Southern Hemisphere should be mentioned among the useful next steps.*

Thank you, yes, I completely agree that it is a pity that I could not expand the evaluation towards Antarctic sea ice. I thought that with the sentence in Line 178/179: "Future efforts should include revisiting the Landsat-1 MSS image archive to check whether there are not more images that can be used, and making use of the existing cloud-free parts of the few existing images as much as possible." I stressed the need to focus on these activities well enough. But I can change this accordingly, by e.g. adding "with a focus on the Antarctic" in the last sentence of the respective paragraph.

*The whole manuscript is written in the first-person style. To the reviewer's impression, this overly emphasizes the role of the author. It appears less appropriate to the reviewer at the many places, where just scientifically agreed standard work is described.*

Indeed I once learned that using "active voice", i.e. using "we" or "I" makes a manuscript more easy to read, enhances the flow, and allows to discriminate better between work that has been done elsewhere and work that has been done in the context of this manuscript. I went through the manuscript, and replaced active voice with passive voice so that the reader's impression improves.

I suggest accepting the manuscript after minor revisions.

*General: ‚sqkm' for square km is an unconventional notation, better adhere the SI system with km².*

Thank you, I changed "sqkm" to "km²" throughout the manuscript.

*L(ine) 40 ‚I did not compute monthly means for months with daily data from 12 or fewer days.': Mention that ESMR only gave data each second day.*

Actually, ESMR indeed gave data every day while SMMR onboard NIMBUS-7 gave data every other day. I hence do not change or add anything here.

*L45 ''I observe differences between the monthly SIE values of up to 2 million sqkm in the Northern Hemisphere and of up  to almost 4 million sqkm in the Southern Hemisphere.': Do you mean between the 10% and 30% threshold values? In that case, the explanation 2 lines later would not hold: it would simply represent the area of SIC between 10% and 30%. Otherwise, specify differences between which quantitites.*

The respective part reads now: "Comparing the monthly SIE values obtained using the two different SIC thresholds, I find differences between 1 and 1.5 million km² in the Northern Hemisphere and between 1 and 3 million km² in the Southern Hemisphere. Only a part of this difference in SIE naturally results from using the two different thresholds specified. A circum-Antarctic band of 25 x 25 km² grid cells located at 60 degrees latitude with SIC values between 15% and 30% would result in a SIE contribution of only about 500 000 km². At least half of the differences in the SIE computed here using a SIC threshold of 15% of 30% is caused by the relatively large retrieval noise over open water, resulting from a less reliable correction of the atmospheric influence, and a larger uncertainty of the tie points used (Kolbe et al., 2024; Tonboe et al., 2025)."

*L 157 equalling -> equaling*

Changed accordingly.

*L 164 'The second one' -> The second deficiency*

Changed accordingly.